# Large variation in anti-SARS-CoV-2 antibody prevalence among essential workers in Geneva, Switzerland

Silvia Stringhini [1,2,3✉], María-Eugenia Zaballa [1], Nick Pullen[1], Carlos de Mestral [1,3], Javier Perez-Saez[4,5], Roxane Dumont[1], Attilio Picazio[1], Francesco Pennacchio [1], Yaron Dibner[1], Sabine Yerly [6], Helene Baysson[1,2], Nicolas Vuilleumier[2,6,7], Jean-François Balavoine[7], Delphine Bachmann[8], Didier Trono [9], Didier Pittet[10], François Chappuis[11], Omar Kherad[12], Laurent Kaiser[6,7,13], Andrew S. Azman [1,4,5], SEROCoV-WORK + Study Group* & Idris Guessous[2,11]

Limited data exist on SARS-CoV-2 infection rates across sectors and occupations, hindering our ability to make rational policy, including vaccination prioritization, to protect workers and limit SARS-CoV-2 spread. Here, we present results from our SEROCoV-WORK+ study, a serosurvey of workers recruited after the first wave of the COVID-19 pandemic in Geneva, Switzerland. We tested workers (May 18—September 18, 2020) from 16 sectors and 32 occupations for anti-SARS-CoV-2 IgG antibodies. Of 10,513 participants, 1026 (9.8%) tested positive. The seropositivity rate ranged from 4.2% in the media sector to 14.3% in the nursing home sector. We found considerable within-sector variability: nursing home (0%–31.4%), homecare (3.9%–12.6%), healthcare (0%–23.5%), public administration (2.6%–24.6%), and public security (0%–16.7%). Seropositivity rates also varied across occupations, from 15.0% among kitchen staff and 14.4% among nurses, to 5.4% among domestic care workers and 2.8% among journalists. Our findings show that seropositivity rates varied widely across sectors, between facilities within sectors, and across occupations, reflecting a higher exposure in certain sectors and occupations.

[1] Unit of Population Epidemiology, Division of Primary Care Medicine, Geneva University Hospitals, Geneva, Switzerland. [2] Department of Health and Community Medicine, Faculty of Medicine, University of Geneva, Geneva, Switzerland. [3] University Center for General Medicine and Public Health, University of Lausanne, Geneva, Switzerland. [4] Department of Epidemiology, Johns Hopkins Bloomberg School of Public Health, Baltimore, MD, USA. [5] Institute of Global Health, Faculty of Medicine, University of Geneva, Geneva, Switzerland. [6] Division of Laboratory Medicine, Geneva University Hospitals, Geneva, Switzerland. [7] Department of Medicine, Faculty of Medicine, University of Geneva, Geneva, Switzerland. [8] Hirslanden Clinique des Grangettes and Hirslanden Clinique La Colline, Geneva, Switzerland. [9] School of Life Sciences, Ecole Polytechnique Fédérale de Lausanne (EPFL), Lausanne, Switzerland. [10] Infection Control Program and World Health Organization Collaborating Center on Patient Safety, Geneva University Hospitals and Faculty of Medicine, Geneva, Switzerland. [11] Division and Department of Primary Care Medicine, Geneva University Hospitals, Geneva, Switzerland. [12] Division of Internal Medicine, Hôpital de la Tour and Faculty of Medicine, Geneva, Switzerland. [13] Geneva Center for Emerging Viral Diseases and Laboratory Virology, Geneva University Hospitals, Geneva, Switzerland. *A list of authors and their affiliations appears at the end of the paper. ✉email: silvia.stringhini@hcuge.ch

As the first wave of the ongoing COVID-19 pandemic swept across the northern hemisphere in the spring of 2020, most countries adopted wide-ranging measures to limit the spread of severe acute respiratory syndrome coronavirus 2 (SARS-CoV-2)[1]. Most measures, however, imposed little or no restrictions on 'essential' workers—whose occupations are deemed indispensable for the provision of crucial services, including healthcare, transportation, food production, social work, among others[2–4]. Depending on the nature of their function, workers may be exposed to infectious, often asymptomatic[5], members of the public and/or colleagues, increasing their risk of infection compared with individuals working from home—and beyond the risk of household transmission[6,7]. Cumulating evidence indicates that healthcare workers in hospitals face an increased risk of infection, although the risk may be department-specific or associated to social rather than professional activities, whilst some studies have found no increased risk[8–13]. Some evidence shows that healthcare workers in nonhospital settings, such as nursing homes, are also at increased risk[14–16]. Less is known about occupations outside healthcare settings, though evidence from RT-PCR testing indicates that public-facing workers such as waiters, social workers, and transport drivers may be at increased risk[17,18]. As the second or third waves of the pandemic spread across the world and mass vaccination starts, there remains an urgent need to better characterize the risk of SARS-CoV-2 infection among workers who are mobilized during lockdowns in order to better guide public health policy to both limit the spread of the virus and protect exposed workers, including in vaccine prioritization.

The canton of Geneva in Switzerland reported its first confirmed COVID-19 case on February 26, 2020, and by April 26, 2021, 54,964 confirmed cases (110.0 per 1000 inhabitants) and 729 deaths had been reported[19]. Lockdown measures—similar to those implemented regionally or nationally across Europe and North America—were imposed on March 16, 2020 from which only essential businesses remained operational and the population was encouraged to stay home; progressive lifting of restrictions took place from April 27, 2020 until June 6, 2020. To understand the SARS-CoV-2 infection risk among workers from key sectors that remained operational during the lockdown, we established the SEROCoV-WORK + serology study, using the same testing procedure as in our SEROCoV-POP population-based seroprevalence study[20,21], whose representative sample can serve as comparison.

## Results

We included 10,513 participants in our analytical sample (55.6% women; mean age [standard deviation]: 43.0 years [10.7]) after excluding 69 participants with missing data or outside the target age range (18–65 years). About 48% had a tertiary education, while 8.3% had only a primary education. Most participants were non-smokers (58.3% versus 25.9% current smokers), 32.0% were overweight and 12.0% were obese. Participants represented 16 activity sectors, ranging from 1668 in healthcare (15.9%), 1185 in transportation (11.3%), and 1102 in nursing homes (10.5%) to 182 in construction (1.7%), 166 in media (1.6%) and 97 in agriculture (0.9%) (Table 1).

Overall, 9.8% of individuals (1026/10513) tested positive for SARS-CoV-2 anti-S1 IgG antibodies (Table 1), roughly the same seropositivity rate (7.9% [95% CrI: 6.8–8.9]) as the general working-age population of Geneva over a similar time period[21]. Older participants (≥50 years) had a lower risk of being seropositive (RR: 0.70; 95% CrI: 0.56–0.83) compared with 18–34-year-old participants. Relative to participants with a doctorate, the risk of being seropositive was lower among participants with

an apprenticeship (RR: 0.54; 95% CrI: 0.35–0.76) and among those with up to secondary education (RR: 0.65; 95% CrI: 0.45–0.90). Compared with non-smokers, smokers and ex-smokers had a reduced risk of being seropositive (RR: 0.45, 95% CrI: 0.30–0.57; RR: 0.72, 95% CrI: 0.55–0.88, respectively).

Seropositivity rate varied widely across sectors, from as high as 14.3% in the nursing home sector to as low as 4.2% in media. Seropositivity rate reached 12.1% in the homecare sector, 11.1% in healthcare, 11.0% in pharmacy and 10.1% in the food industry. Relative to healthcare sector participants, those in the public security sector (RR: 0.53; 95% CrI: 0.22–0.93) and early childhood education sector (RR: 0.37; 95% CrI: 0.09–0.84) were at lower risk of being seropositive. Participants who were fully confined during the lockdown were more likely to be seropositive (RR: 1.27; 95% CrI: 1.03–1.56), compared with mobilized participants. In line with this finding, participants with at least one out-of-work exposure to a confirmed COVID-19 case were more likely to be seropositive (RR: 2.29; 95% CrI: 1.93–2.74) compared with their counterparts without any exposure.

A large heterogeneity in seropositivity rate was observed across participating facilities within every sector (Fig. 1). For example, it ranged from 0 to 31.4% across 21 nursing homes, and from 2.6 to 24.6% across 8 public administration facilities. Furthermore, occupations within facilities and sectors also showed varying degrees of seropositivity rate. Across the 32 occupations, seropositivity rate was as high as 15.0% among kitchen staff and 14.4% among nurses/nurse assistants, and as low as 5.4% among domestic care workers and 2.8% among journalists (Fig. 2).

Sensitivity analyses excluding participants who were confined during the lockdown yielded similar results (Supplementary figs. 1–2; Supplementary table 3). Disaggregating seropositivity rates by out-of-work exposure showed that participants with at least one out-of-work exposure to a confirmed COVID-19 case generally had twice the seropositivity rate compared with their same-sector or same-occupation counterparts with no out-of-work exposure (Supplementary tables 4–5). Further adjusting relative risk estimates for known out-of-work exposures, however, showed no appreciable change from the main results (Supplementary table 6).

## Discussion

In this large sample of workers from 16 sectors who were mobilized during the spring 2020 Swiss lockdown, we observed that the proportion of workers having developed anti-SARS-CoV-2 antibodies after the first COVID-19 wave varied widely across sectors, across facilities within sectors and across occupations within sectors. With few important exceptions, our findings do not show a pattern of increased risk of SARS-CoV-2 infection among sectors and occupations of workers who were mobilized during the lockdown, the overall seropositivity rate of this sample being only slightly higher than that of the general working-age population during the spring 2020[21]. Yet, there was considerable variability across sectors and occupations.

While nursing home workers exhibited the highest seropositivity rate relative to that of the working-age population, reflecting evidence from Spain[14], Sweden[15], and the UK[16], it differed widely across nursing home facilities; this intra-sector variability, which was observed in almost all sectors, may reflect overdispersion, a well-known characteristic of SARS-CoV-2 transmission dynamics[22,23]. Whether the degree of adherence to preventive measures within facilities and in private life may also help explain this heterogeneity will be elucidated in further studies. It is also possible that most infections had occurred before strict measures were implemented. Healthcare sector workers showed a generally higher seropositivity rate than the

**Table 1 Sample description and prevalence of anti-SARS-CoV-2 IgG antibodies, SEROCoV-WORK + study, May–September 2020, Geneva, Switzerland.**

|  | N (%) | Seropositive | Relative risk (95% CrI) |
|---|---|---|---|
| Total | 10513 | 1026 (9.8) | – |
| Women | 5848 (55.6) | 598 (10.2) | 1.00 (ref.) |
| Men | 4665 (44.4) | 428 (9.2) | 0.88 (0.76–1.01) |
| Age group, years |  |  |  |
| 18–34 | 2678 (25.5) | 291 (10.9) | 1.00 (ref.) |
| 35–49 | 4524 (43.0) | 467 (10.3) | 0.94 (0.80–1.10) |
| 50–65 | 3311 (31.5) | 268 (8.1) | 0.71 (0.57–0.85) |
| Educational level[a] |  |  |  |
| Mandatory education | 867 (8.3) | 87 (10.0) | 0.75 (0.50–1.07) |
| Apprenticeship | 2216 (21.1) | 175 (9.7) | 0.54 (0.35–0.76) |
| Secondary education | 2098 (20.0) | 194 (9.3) | 0.65 (0.45–0.90) |
| University | 4626 (44.0) | 491 (10.6) | 0.77 (0.56–1.04) |
| Doctorate | 413 (3.9) | 52 (12.6) | 1.00 (ref.) |
| Other | 289 (2.7) | 27 (9.3) | – |
| Activity sector[b] |  |  |  |
| Healthcare | 1668 (15.9) | 185 (11.1) | 1.00 (ref.) |
| Transportation | 1185 (11.3) | 99 (8.4) | 0.66 (0.31–1.18) |
| Nursing homes | 1102 (10.5) | 157 (14.3) | 1.24 (0.67–2.07) |
| Public administration | 1056 (10.0) | 94 (8.9) | 0.91 (0.41–1.74) |
| Public security | 1055 (10.0) | 83 (7.9) | 0.53 (0.22–0.93) |
| Food industry | 755 (7.2) | 76 (10.1) | 0.92 (0.46–1.67) |
| Homecare | 753 (7.2) | 91 (12.1) | 0.99 (0.33–2.06) |
| Social work | 734 (7.0) | 62 (8.5) | 0.68 (0.27–1.33) |
| Financial services | 528 (5.0) | 65 (12.3) | 1.00 (0.42–1.93) |
| International organizations | 425 (4.0) | 24 (5.7) | 0.47 (0.10–1.40) |
| Early childhood education | 259 (2.5) | 15 (5.8) | 0.37 (0.09–0.84) |
| Pharmacy | 254 (2.4) | 28 (11.0) | 0.86 (0.37–1.58) |
| Construction | 182 (1.7) | 11 (6.0) | 0.53 (0.10–1.52) |
| Media | 166 (1.6) | 7 (4.2) | 0.55 (0.07–2.02) |
| Agriculture | 97 (0.9) | 8 (8.3) | 0.65 (0.13–1.70) |
| Other | 294 (2.8) | 21 (7.1) | 0.54 (0.17–1.08) |
| Confinement status[c] |  |  |  |
| Mobilized/partially confined | 9430 (89.9) | 899 (9.5) | 1.00 (ref.) |
| Fully confined | 1065 (10.1) | 125 (11.7) | 1.27 (1.03–1.56) |
| Out-of-work exposure[d] |  |  |  |
| 0 | 9129 (87.0) | 775 (8.5) | 1.00 (ref.) |
| ≥1 | 1365 (13.0) | 248 (18.2) | 2.29 (1.93–2.74) |
| Smoking |  |  |  |
| Non-smoker | 6122 (58.3) | 709 (11.6) | 1.00 (ref.) |
| Ex-smoker | 1664 (15.8) | 143 (8.6) | 0.72 (0.55–0.88) |
| Smoker | 2718 (25.9) | 172 (6.3) | 0.45 (0.30–0.57) |
| BMI group |  |  |  |
| 18–24.9 | 5747 (56.1) | 585 (10.2) | 1.00 (ref.) |
| 25–29.9 | 3278 (32.0) | 306 (9.3) | 0.96 (0.81–1.13) |
| ≥30 | 1227 (12.0) | 115 (9.4) | 0.96 (0.75–1.19) |
| Chronic conditions[e] |  |  |  |
| None | 9282 (88.3) | 903 (9.7) | 1.00 (ref.) |
| 1 | 1062 (10.1) | 106 (10.0) | 1.11 (0.87–1.38) |
| ≥2 | 167 (1.6) | 17 (10.2) | 1.19 (0.62–1.89) |

Results are N (%), or as stated. Relative risks (95% credible interval) are from Bayesian logistic regression models, and are adjusted for test performance, age and sex. These pertain to the reference individual (Female, 18–34). 69 participants were excluded due to missing serology or sociodemographic data, or due to being outside target age range of 18–65 years. All data beside serology are self-reported.
[a]Each category indicates the highest level of education attained by participant; mandatory education indicates 15 years of primary education as the highest obtained degree.
[b]See Supplementary Table 1 for details.
[c]Fully confined at home between March and May 2020; otherwise partially confined or fully mobilized at work.
[d]Out-of-work exposure to confirmed COVID-19 cases.
[e]Chronic conditions: hypertension, diabetes, cardiovascular disease, respiratory disease, cancer.

general population, consistent with extensive evidence from many countries[8–10,14,18].

Among occupations, the highest proportion of seropositive workers was observed among kitchen staff, who worked primarily in nursing homes; this may reflect the increased infection risk that is present when customers/staff fail to practice appropriate hand hygiene, social distancing, and mask wearing when indicated—measures that are difficult to follow while eating and drinking indoors[24–26]. Nurses also exhibited higher seropositivity than the general population, a pattern consistently reported elsewhere[8,9,11–13], and likely reflective of the extended cumulative exposure time to SARS-CoV-2 they have as front-line patient-facing workers[22,27]; importantly, contact tracing data have shown that the majority of nurses working in the main healthcare institution in Geneva who were infected with SARS-CoV-2 became infected during social interactions or transportation to

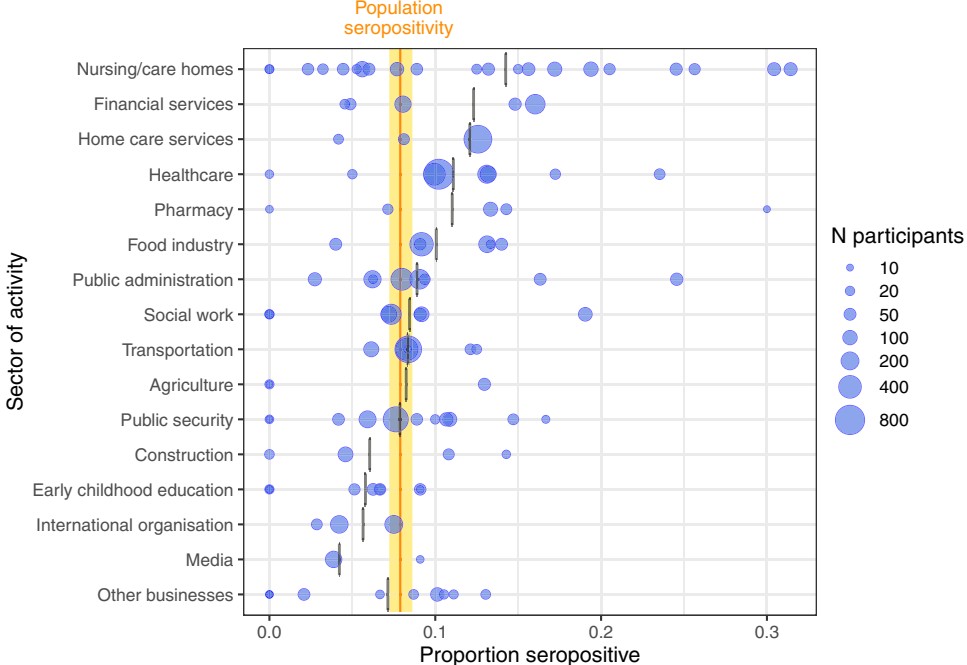

**Fig. 1 Prevalence of anti-SARS-CoV-2 IgG antibodies by activity sector, SEROCoV-WORK + study, May–September 2020, Geneva, Switzerland.**
Sample size: 10,513 participants, 1026 of which were seropositive. Blue dots represent proportion of seropositive participants per company/workplace facility. Dot size indicates number of employees participating. Darker dots indicate more than one facility with same or very similar seropositivity rate. Vertical orange bar and yellow area indicate general working-age population seropositivity rate and 95% binomial confidence interval, respectively, from SEROCoV-POP study[20,21]. Small gray vertical bars show the proportion positive of all participants per sector. Facilities with <10 participants are not shown as dots, but these participants are included in the sector average.

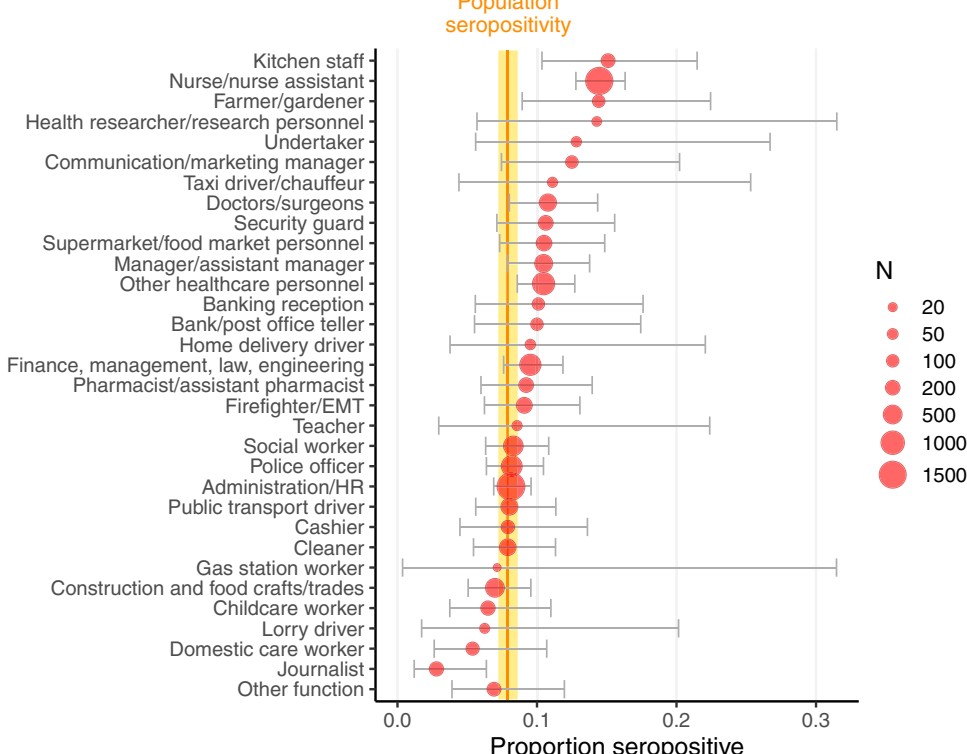

**Fig. 2 Prevalence of anti-SARS-CoV-2 IgG antibodies by occupation, SEROCoV-WORK + study, May–September 2020, Geneva, Switzerland.** Sample size: 10,513 participants, 1026 of which were seropositive. Red dots indicate mean seropositivity rate for each occupation, while horizontal gray lines represent 95% binomial confidence intervals. Dot size indicates number of employees with that occupation. Vertical orange bar and yellow area indicate general working-age population seropositivity rate and 95% binomial confidence interval, respectively, from SEROCoV-POP study[20,21].

and from work[28]. Still, as mass vaccination campaigns are being planned and implemented worldwide, our results strengthen the evidence to prioritize access to healthcare workers, particularly nurses, as well as employees of nursing homes, including kitchen staff.

While we did not find a clear trend in the association between educational level and risk of infection as reported in other countries[29–31], we found that those with a doctorate degree were at highest risk, reflecting our previous findings in the Geneva population[21]; this risk is likely linked to the type of occupations held by these individuals, most of whom worked in the healthcare sector in our sample. The observed higher risk of being seropositive among fully confined participants may be due to the relatively high risk of transmission within the household when a single household member was infected during the first wave[6,7], or due to having been infected before or after the mandatory homeworking period. Participants with out-of-work exposure to confirmed cases of COVID-19 had twice the risk of infection compared with their non-exposed counterparts in the same-sector and occupation. This out-of-work exposure is likely determined by several socioeconomic and demographic factors, which in turn are also associated with occupation. While our results showed that out-of-work exposure was associated with higher seropositivity rates, it only partly explained the wide heterogeneity observed across sectors and occupations. The observed reduced risk of being seropositive in ex-smokers and current smokers reflected our previous population-based findings[21]; this may be due to potential interaction between the SARS-CoV-2 spike protein and nicotinic acetylcholine receptors[32], heightened preventive practices and risk aversion among smokers given their increased risk of viral and bacterial respiratory diseases[33,34], or residual confounding. Importantly, while contrasting findings have been reported about the association between smoking and SARS-CoV-2 infection risk, extensive evidence indicates that smoking is associated with increased risk of severe COVID-19[35,36].

Our findings provide an important picture of SARS-CoV-2 infection in a large and diverse sample of workers considered to be at higher-risk of exposure. However, we acknowledge some important limitations. Participants were selected from a list of potentially mobilized facilities in the canton of Geneva—our sample may thus not be representative of the overall population of mobilized workers. Despite the 4-month recruitment period, during which weekly reported infections remained low in the canton of Geneva (Supplementary Fig. 3), time of participation in the study was not related with seropositivity rates (Supplementary Fig. 4). Importantly, the time frame captured in our data included periods before and after population-wide government-mandated preventive measures were implemented—including heterogeneous sector-specific measures and a lockdown, followed by progressive lifting of lockdown measures. It is likely that the inter- and intra-sectoral variability in seropositivity rate could be partly explained by preventive measures. However, our data do not allow to account for the potential impact of preventive measures. Finally, although we accounted for self-reported out-of-work exposure to confirmed cases of COVID-19, unknown exposure to SARS-CoV-2 outside the work setting, such as in transportation to and from work, may have also played a role.

In conclusion, we conducted a serosurvey among workers in 16 sectors deemed as essential for the smooth functioning of society during lockdowns. With the exclusion of the nursing home, homecare and healthcare sectors, as well as nurses/nurse assistants, and kitchen employees within facilities, we found little evidence to support that workers in sectors that were not confined during the initial COVID-19 lockdown faced a greater risk of contracting SARS-CoV-2 during the first wave than the general working-age population. Importantly, seropositivity rates differed widely across sectors, between facilities within sectors, and across occupations.

## Methods

As there was no list of workers mobilized during the pandemic in the canton of Geneva (a state of Switzerland with about 500,000 inhabitants) from which we could draw our sample, we selected public and private companies and institutions—hereafter facilities—that were potentially mobilized (see Supplementary Note 1 for further selection and recruitment information). Facilities were eligible for participation if they were located in the canton of Geneva and had remained mostly operational with on-site staff activity during the lockdown. Participating facilities in turn invited their employees to participate on a voluntary basis. The mean participation rate per facility was 45% (median 41%). All participants gave written informed consent, completed a questionnaire and provided a venous blood sample. Samples in the SEROCoV-WORK+ study were collected from May 18 until September 18, 2020, while samples in the population-based survey—which data are used here for comparison—were collected from April 6 until June 30, 2020[21]. The recruitment phase of both studies was completed before the beginning of the second pandemic wave, and SARS-CoV-2 circulation was low in Geneva until the end of September 2020[19]. The Cantonal Research Ethics Commission of Geneva, Switzerland, approved the study (project number 2020-00881).

We assessed anti-SARS-CoV-2 IgG antibodies using a commercially available ELISA (Euroimmun; Lübeck, Germany #EI 2606-9601G) targeting the S1 domain of the virus spike protein, using the manufacturer's recommended cut-off of $\geq 1.1$ for seropositivity[37]. We standardized occupations into 16 sectors and 32 occupation groups (Supplementary Tables 1–2). To estimate the relative risk (posterior mean and 95% central credible interval) of being seropositive across different groups of participants, we fit Bayesian regression models that accounted for age, sex, and test performance including, when appropriate, random effects for facilities. We implemented these models in the Stan probabilistic programming language, using the *rstan* package[38]. We ran 5000 samples (four chains of 1500 iterations, each with 250 warmup iterations discarded), assessing convergence visually and using *shinystan* diagnostics checks[39]. In sensitivity analyses, we excluded participants who were fully confined during the lockdown, evaluated seropositivity rate according to out-of-work exposure to confirmed COVID-19 cases, and incorporated out-of-work exposure as a covariate in the main model. Full details of the model are in the Supplementary Information. For comparison, all figures include the seropositivity rate from the SEROCoV-POP representative sample of the Geneva working-age population[20,21].

**Reporting summary**. Further information on research design is available in the Nature Research Reporting Summary linked to this article.

## Data availability

Participants' informed consent does not authorize data, even coded, to be immediately available. It does allow, however, for the data to be made available to the scientific community upon submission of a data request application to the investigators board via the corresponding author. Virologically-confirmed SARS-CoV-2 infection data from the Canton of Geneva: https://infocovid.smc.unige.ch/. Biological material can be reused for further studies upon approval by the cantonal ethics commission.

## Code availability

Analytical code is available from https://github.com/UEP-HUG/SEROCoV-WORK-Public.

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

## Acknowledgements

We thank the Hôpital de la Tour, the Hirslanden Clinique des Grangettes, and the Hirslanden Clinique de la Colline for participating as testing centers for the study, and the Geneva Chamber of Commerce, Industry, and Services (CCIG) for contributing to the recruitment of facilities. We are grateful to the members of the SEROCoV-WORK + Integrity Board, for dedicating time to evaluate the submitted lists of potential participating facilities. We thank the staff in all participating facilities, as well as to all participants, whose contributions were invaluable and integral to the study. This study was funded by the Private Foundation of the Geneva University Hospitals, the Fondation des Grangettes and the Center for Emerging Viral Diseases.

## Author contributions

I.G. and S.S. initiated the study. I.G., S.S., L.K., O.K., D.P., F.C., J-F.B., H.B., D.T., D.B. designed the study. S.S., M-E.Z., I.G., O.K., D.B., F.C. acquired the data; S.S., I.G., M-E.Z., P.N., CdM., A.S.A., J.P-S. determined statistical analyses. M-E.Z., Y.D., F.P., A.P. and R.D. conducted data analyses during recruitment. S.Y., N.V., and L.K. conducted laboratory analyses. N.P. and J.P-S. conducted main statistical analyses and created graphics; C.dM., M-E.Z., N.P. wrote the paper; S.S., M-E.Z., N.P., C.dM., J.P-S., R.D., A.P., F.P., Y.D., S.Y., H.B., N.V., J-F. B., D.B., D.T., D.P., F.C., O.K., L.K., A.S.A., and I.G. commented on drafts of paper and approved its final version.

## Competing interests

The authors declare no competing interests.

## Additional information

**SEROCoV-WORK + Study Group**

Victoria Alber[9], Isabelle Arm-Vernez[14], Andrew S. Azman [1,4,5], Delphine Bachmann[8,15], Donatien Bachmann[1,8], Stéphanie Baggio[16], Jean-François Balavoine[7], Gil Barbosa Monteiro[14], Hélène Baysson[1,2], Patrick Bleich[1], Isabelle Boissel[15], François Chappuis[11], Prune Collombet[1], Delphine Courvoisier[17], Philippine Couson[8], Alioucha Davidovic[1], Clement Deiri[8], Divina Del Rio[15], Carlos de Mestral[1,3], David De Ridder[2,11], Yaron Dibner[1], Paola D'ippolito[1], Joséphine Duc[1], Roxane Dumont[1], Isabella Eckerle[13,18,19], Nacira El Merjani[1], Gwennaelle Ferniot[17,20], Antonie Flahault[5], Natalie Francioli[1], Marion Frangville[1,21], Carine Garande[17], Laurent Gétaz[16,22], Pamela Giraldo[20], Fanny Golaz[6], Julie Guérin[1], Idris Guessous[2,11], Ludivine Haboury[1], Séverine Harnal[1], Victoria Javet[8], Laurent Kaiser[6,7,13,18,19], Omar Kherad[12], Amélie Laboulais[20], Gaëlle Lamour[1], Xavier Lefebvre[20], Pierre Lescuyer[6], Andrea Jutta Loizeau[1], Fanny-Blanche Lombard[1], Elsa Lorthe[1], Chantal Martinez[1], Kourosh Massiha[1], Ludovic Metral-Boffod[6], Benjamin Meyer[23], Khaled Mostaguir[24], Mayssam Nehme[11], Natacha Noël[1], Nicolas Oederlin[1], Francesco Pennacchio [1], Javier Perez-Saez[4,5], Dusan Petrovic[1,3], Attilio Picazio[1], Didier Pittet[10], Giovanni Piumatti[1], Jane Portier[11], Géraldine Poulain[6], Caroline Pugin[1], Nick Pullen[1], Barinjaka Rakotomiaramanana[1], Zo Francia Randrianandrasana[1,15], Aude Richard[1], Viviane Richard[1], Sabina Rodriguez-Velazquez[1], Lilas Salzmann-Bellard[1], Silvia Stringhini [1,2,3✉], Leonard Thorens[8], Simon Torroni[15], Didier Trono [9], David Vidonne[8], Guillemette Violot[21], Nicolas Vuilleumier[2,6,7], Zoé Waldmann[1], Manon Will[1], Ania Wisniak[1]Sabine Yerly [6] & María-Eugenia Zaballa [1]

[14]Laboratory of Virology, Geneva University Hospitals, Geneva, Switzerland. [15]Hirslanden Clinique La Colline, Geneva, Switzerland. [16]Division of Prison Health, Geneva University Hospitals, Geneva, Switzerland. [17]General Directorate of Health, Geneva, Switzerland. [18]Department of Microbiology and Molecular Medicine, Faculty of Medicine, University of Geneva, Geneva, Switzerland. [19]Division of Infectious Diseases, Geneva University Hospitals, Geneva, Switzerland. [20]Hôpital de la Tour, Geneva, Switzerland. [21]Communication Directorate, Geneva University Hospitals, Geneva, Switzerland. [22]Division of Tropical and Humanitarian Medicine, Geneva University Hospitals, Geneva, Switzerland. [23]Department of Pathology and Immunology, Center for Vaccinology, Faculty of Medicine, University of Geneva, Geneva, Switzerland. [24]Clinical Research Center, Geneva University Hospitals, Geneva, Switzerland.

