## [Peer Review File · Nature Communications]

REVIEWER COMMENTS

Reviewer #1 (Remarks to the Author):

Strinhini and colleagues present seroprevalence across various essential sectors and occupations, assumed to be at increased risk of transmission, and relate these prevalences with those found in a large serological survey of the Geneva working age population (SEROCoV-POP). Main findings are that workers in sectors and occupations not confined to lock-down are generally not at greater risk of contracting SARS-CoV-2 than the general working population. With the exception of healthcare sectors. The paper is well-written, the over-all study design is sound and the sample size is large, generating important data.

Minor comments:

The sector seroprevalences are compared with that of the general population, found in a separate study. Were samples collected during the same time periods in both these studies?

As pointed out by the authors, a possible selection bias may be due to the voluntary enrolment. Was the enrolment procedure similar in the SEROCOVID-POP study?

It is interesting that there are such large variations within sectors. This may be the most interesting finding, and suggests that differences in prevention strategies employed within these sectors may be of importance. The authors imply that this will be looked into. To increase the value of this paper, I would strongly suggest to add this information.

I would suggest a deeper discussion into how this data might influence vaccine strategies? Are healthcare workers already being prioritized in Geneva?

Fig 1- Why are there darker dots within some dots?

Reviewer #2 (Remarks to the Author):

This is a concise, well written report of a serosurvey of approximately 10,000 essential workers in Geneva following the first COVID-19 surge with the goal of understanding the SARS-CoV-2 infection risk in workplaces that remained open during the lockdown. The investigators were able to piggy-back on previous work done with a similar population based survey. The modeling to estimate the relative risk of seropositivity accounted for age, sex and test performance. The conclusion that there is little evidence to support that workers in sectors that were not confined during the initial lockdown faced greater risk of contracting SARS-CoV-2 infection is supported by the data presented. However, there does not appear to be accounting for exposures outside the workplace, including known exposures to a COVID positive contact that might not be uniform across sectors, making it difficult to compare the relative risk of SARS-CoV-2 seropositivity from workplace exposure across the various sectors.

Interestingly, the previous work done by this group suggests that the disparities/socioeconomic factors that markedly impact COVID risk in other countries may not be as relevant in Geneva. That said, convincing the reader that factors unrelated to workplace exposure have been accounted for is important.

Introduction

P 3 lines 55-9 and lines 62-4 The investigators posit that a better understanding of the risk of infection in essential workers may help guide public health policy including vaccine prioritization. The authors then mention that lockdown measures were imposed on March 16 in Geneva. If the first wave occurred before universal masking and other mitigations efforts, and this serosurvey reflected infection risk prior to these interventions, will this serosurvey results actually be able to help us understand ongoing infection risk and help guide policy?

Methods

What was the denominator (number of employees) at the companies/institutions engaged for this

survey? How do we know that the sample was representative of those working in the various sectors?

Supplemental materials state that there were 3 testing sites and that, for some companies with critical constraints, on-site testing was offered. The limited testing sites and selective on-site testing may have biased the sample. Were attempts made to control for this bias?

The supplemental material includes a description of the survey completed by all participants at time of blood draw. This survey included "risk behaviors". Please describe these risk behaviors. Were they included in the analysis?

Was exposure to a person with known COVID-19, particularly outside of the work place, asked in the questionnaire? The findings that those "confined" participants had a higher observed risk of being seropositive is evidence that this household exposure is important.

Discussion

Limitation section should be expanded to include discussion of the possibility of exposure outside of the work place and that this exposure may vary by job type and demographic factors.

Limitations section should also mention that the period of exposure (early in the pandemic) included periods with and without mitigation strategies such as social distancing and masking making it difficult to understand what occurred before or after these interventions.

Response to reviewers' comments

Reviewer #1

Stringhini and colleagues present seroprevalence across various essential sectors and occupations, assumed to be at increased risk of transmission, and relate these prevalences with those found in a large serological survey of the Geneva working age population (SEROCoV-POP). Main findings are that workers in sectors and occupations not confined to lock-down are generally not at greater risk of contracting SARS-CoV-2 than the general working population. With the exception of healthcare sectors. The paper is well-written, the over-all study design is sound and the sample size is large, generating important data.

Our response: We thank the Reviewer for this positive comment.

The sector seroprevalences are compared with that of the general population, found in a separate study. Were samples collected during the same time periods in both these studies?

Our response: Samples in the population-based SEROCov-POP study were collected from April 6 until June 30, 2020, while those of SEROCov-WORK+ study were collected from May 18 until September 18, 2020—there was an overlap of two months between the two studies. The recruitment phase of both studies was completed before the beginning of the second pandemic wave, and SARS-CoV-2 circulation was low in Geneva until the end of September 2020. To clarify the samples' collection time period, we have added the following sentence in the methods section (page 8, paragraph 2):

“Samples in the SEROCov-WORK+ study were collected from May 18 until September 18, 2020, while samples in the population-based survey—which data are used here for comparison—were collected from April 6 until June 30, 2020. The recruitment phase of both studies was completed before the beginning of the second pandemic wave, and SARS-CoV-2 circulation was low in Geneva until the end of September 2020.¹”

As pointed out by the authors, a possible selection bias may be due to the voluntary enrolment. Was the enrolment procedure similar in the SEROCov-POP study?

Our response: The sampling and enrolment procedure differed between the two studies: the population-based survey invited participants from a representative sample of the adult population, while SEROCov-WORK+ invited participants from selected facilities as described in Supplementary table 1. However, once selected either as individual participants or as employees of a company, individual participation in both studies was on a voluntary basis, and the bias was likely to be similar in the two groups. Nevertheless, to acknowledge this important limitation in our recruitment method, we have rephrased the relevant section in the limitations as follows (page 7, paragraph 2):

“Our findings provide an important picture of SARS-CoV-2 infection in a large and diverse sample of workers considered to be at higher-risk of exposure. However, we acknowledge some important limitations. Participants were selected from a list of

potentially mobilized facilities in the canton of Geneva—our sample may thus not be representative of the overall population of mobilized workers.”

It is interesting that there are such large variations within sectors. This may be the most interesting finding, and suggests that differences in prevention strategies employed within these sectors may be of importance. The authors imply that this will be looked into. To increase the value of this paper, I would strongly suggest to add this information.

Our response: We agree that prevention strategies adopted in different companies may explain part of the variability observed within sectors. However, analysis of these data is limited by the following factors:

- 1) Preventive measures were introduced progressively over time, and our recruitment lasted several months (May-September, 2020). While our questionnaire contained several items about preventive measures currently in place, unfortunately we did not ask about measures in place at the time of the first wave. Therefore, the variability of preventive measures within sectors are likely strongly influenced/confounded by the time of recruitment.
- 2) In the early stages of the pandemic, when a large proportion of participants are likely to have become infected, few government-mandated preventive measures had been adopted, and there was enormous heterogeneity across and within different sectors of society in preventive measures that were being adopted. Conversely, sectors that were more likely to have introduced preventive measures early on, such as healthcare settings mandating mask wearing and hand washing, were also those where exposure to SARS-CoV-2 was higher.
- 3) SARS-CoV-2 spreads in clusters after introduction into a specific setting, so it would be important to account for potential clusters within facilities while modeling the impact of preventive measures. Our data unfortunately does not allow to account for this.

In light of these limiting factors, we decided to present a short report describing the evidence regarding the overall seropositivity rate in the “exposed” vs the general population, differences between sectors and occupations, as well as heterogeneity within sectors. An analysis dedicated to the understanding of this heterogeneity would require to complement the data that we have with qualitative information, which we plan to obtain in the next few months in specific settings such as nursing homes, via another study specifically addressing the complex issues of the role of preventive measures.

To acknowledge our inability to assess the role of preventive measures, we have added the following to the limitations section (also addressing comments from Reviewer 2; page 7, paragraph 2):

“Importantly, the time frame captured in our data included periods before and after population-wide government-mandated preventive measures were implemented—including heterogeneous sector-specific measures and a lockdown, followed by progressive lifting of lockdown measures. It is likely that the inter- and intra-sectoral variability in seropositivity rate could be partly explained by preventive measures.

However, our data do not allow to account for the potential impact of preventive measures.”

I would suggest a deeper discussion into how this data might influence vaccine strategies? Are healthcare workers already being prioritized in Geneva?

Our response: We thank the Reviewer for this suggestion. Indeed, access to vaccination is currently prioritized for healthcare workers in Geneva, including those working in nursing homes (<https://www.ge.ch/en/getting-vaccinated-against-covid-19/priority-groups-vaccination>). Our results strengthen the evidence for such current recommendations. We have added the following sentence to the discussion (page 6, paragraph 2):

“As mass vaccination campaigns are being planned and implemented worldwide, our results strengthen the evidence to prioritize access to healthcare workers, particularly nurses, as well as employees of nursing homes, including kitchen staff.”

Fig 1- Why are there darker dots within some dots?

Our response: Some dots appear to have darker dots because the same seropositivity rate was observed for more than one facility. To clarify this, we have added the following sentence in the figure legend (Figure 1 and Supplementary Figure 1):

“Darker dots indicate more than one facility with same or very similar seropositivity rate.”

Reviewer #2

This is a concise, well written report of a serosurvey of approximately 10,000 essential workers in Geneva following the first COVID-19 surge with the goal of understanding the SARS-CoV-2 infection risk in workplaces that remained open during the lockdown. The investigators were able to piggy-back on previous work done with a similar population-based survey. The modeling to estimate the relative risk of seropositivity accounted for age, sex and test performance. The conclusion that there is little evidence to support that workers in sectors that were not confined during the initial lockdown faced greater risk of contracting SARS-CoV-2 infection is supported by the data presented. However, there does not appear to be accounting for exposures outside the workplace, including known exposures to a COVID positive contact that might not be uniform across sectors, making it difficult to compare the relative risk of SARS-CoV-2 seropositivity from workplace exposure across the various sectors.

Interestingly, the previous work done by this group suggests that the disparities/socioeconomic factors that markedly impact COVID risk in other countries may not be as relevant in Geneva. That said, convincing the reader that factors unrelated to workplace exposure have been accounted for is important.

Our response: We thank the Reviewer for these encouraging comments. We agree that information on out-of-work exposure should be reported. We have now added out-of-work exposure to the main table (Table 1), as well as in the supplementary material, showing the seropositivity rate by sector and occupation disaggregated by

presence or absence of out-of-work exposure to confirmed cases of COVID-19 (Supplementary figures 5-6). We found that seropositivity rates were in general twice as high among participants with at least one out-of-work exposure compared with participants with no out-of-work exposure within the same sector or occupation. We have now added the following sentence in the results (page 5, paragraph 1):

“... In line with this finding, participants with at least one out-of-work exposure to a confirmed COVID-19 case were more likely to be seropositive (2.29; 95% CrI: 1.93-2.74) compared with their counterparts without any exposure.”

We have also added the following sentence to the discussion (page 7, paragraph 1):

“Participants with out-of-work exposure to confirmed cases of COVID-19 had twice the risk of infection compared with their non-exposed counterparts in the same sector and occupation. This out-of-work exposure is likely determined by several socioeconomic and demographic factors, which in turn are also associated with occupation. While our results showed that out-of-work exposure was associated with higher seropositivity rates, it only partly explained the wide heterogeneity observed across sectors and occupations.”

Regarding socioeconomic indicators, in a recent population-based work of ours (<https://www.medrxiv.org/content/10.1101/2020.12.16.20248180v1>), we indeed found no evidence that socioeconomic indicators were associated with COVID-19 risk. We also presented seropositivity rates by education in this report (Table 1), confirming findings from our population-based survey. We have now added a sentence in the results section (page 4, paragraph 3):

“Relative to participants with a doctorate, the risk of being seropositive was lower among participants with an apprenticeship (RR: 0.54; 95% CrI: 0.35-0.76) and among those with up to secondary education (RR: 0.65; 95% CrI: 0.45-0.90).”

We have also added the following sentence to the discussion (page 6, paragraph 3):

“While we did not find a clear trend in the association between educational level and risk of infection as reported in other countries,²⁻⁴ we found that those with a doctorate degree were at highest risk, reflecting our previous findings in the Geneva population;⁵ this risk is likely linked to the type of occupations held by these individuals, most of whom worked in the healthcare sector in our sample.”

Introduction

P 3 lines 55-9 and lines 62-4 The investigators posit that a better understanding of the risk of infection in essential workers may help guide public health policy including vaccine prioritization. The authors then mention that lockdown measures were imposed on March 16 in Geneva. If the first wave occurred before universal masking and other mitigations efforts, and this serosurvey reflected infection risk prior to these interventions, will this serosurvey results actually be able to help us understand ongoing infection risk and help guide policy?

Our response: We thank the Reviewer for this important question. As seen in our now modified **Supplementary Figure 3** (also shown below), the first wave of the COVID-

19 pandemic was only reaching its peak when lockdown measures were implemented in Geneva. Additionally, while universal masking was indeed implemented later on, prevention measures started being implemented, albeit very heterogeneously across different public and private spaces, before the government-mandated lockdown began. Therefore, our data likely reflect infection risk after some prevention measures began being implemented. Nevertheless, to acknowledge this important limitation, we have added the following sentence to the limitations section (also in response other comments by Reviewer 1; page 7, paragraph 2):

“Importantly, the time frame captured in our data included periods before and after population-wide government-mandated preventive measures were implemented—including heterogeneous sector-specific measures and a lockdown, followed by progressive lifting of lockdown measures. It is likely that the inter- and intra-sectoral variability in seropositivity rate could be partly explained by preventive measures. However, our data do not allow to account for the potential impact of preventive measures.”

Background shading indicates government-mandated preventive measures in place:

- Darker blue** Lockdown imposed in canton Geneva on March 16, 2020.
- Middle blue** First phase of easing lockdown measures began on April 27, 2020.
- Lighter blue** Second phase of easing lockdown measures on began on May 11, 2020.
- No shade** All lockdown measures lifted in canton Geneva on June 6, 2020.
- Lighter green** Federally-mandated mask wearing in public transport began on July 6, 2020 (ongoing as of March 10, 2021).
- Darker green** Mask wearing inside stores became mandatory in canton Geneva on July 28, 2020 (ongoing as of March 10, 2021).

Methods

What was the denominator (number of employees) at the companies/institutions engaged for this survey? How do we know that the sample was representative of those working in the various sectors?

Our response: We thank the Reviewer for this question. Eligible companies and institutions were those located in the canton of Geneva and that remained mostly operational with on-site activity during the spring 2020 lockdown. Invitation to

participate in the study was based exclusively on these criteria and only limited by logistics constraints (for instance, inability to reach the representative or contact person of a company), and no systematic bias was done in the selection of participating facilities in the various sectors. Only 14% of the companies/institutions that were directly contacted actively refused to participate and about 20% did not respond to our direct invitation. This is not surprising given the context of a rapidly evolving pandemic and lockdown measures. Within companies that accepted to participate in the study, mean participation rate (participants/invited employees) was 45% (median 41%).

To clarify our recruitment method, we have now rephrased the relevant section in the methods (page 8, paragraph 2):

“As there was no list of workers mobilized during the pandemic in the canton of Geneva (a state of Switzerland with about 500,000 inhabitants) from which we could draw our sample, we selected public and private companies and institutions—hereafter facilities—that were potentially mobilized (see Supplementary Table 1 for further selection and recruitment information). Facilities were eligible for participation if they were located in the canton of Geneva and had remained mostly operational with on-site staff activity during the lockdown. Participating facilities in turn invited their employees to participate on a voluntary basis. The mean participation rate per facility was 45% (median 41%).”

We have now also rephrased relevant sentence in the limitations section (page 7, paragraph 2):

“Participants were selected from a list of potentially mobilized facilities in the canton of Geneva—our sample may thus not be representative of the overall population of mobilized workers.”

Supplemental materials state that there were 3 testing sites and that, for some companies with critical constraints, on-site testing was offered. The limited testing sites and selective on-site testing may have biased the sample. Were attempts made to control for this bias?

Our response: We thank the Reviewer for raising this important question. We had the same concern when starting the study and despite the fact that the canton of Geneva is a very dense and small geographic area, we proposed multiple testing sites to mitigate the risk of bias. The three testing sites were distantly located across the main urban area of the Geneva canton allowing most employees to find one study testing center at a reasonable distance from their home or work address. We did not find any indication for bias when looking at the obtained results. For example, within the nursing home sector, where participants from 14 facilities were invited to come to one of the three testing centers while participants from 7 other facilities were tested on site, findings show that facilities in both groups were similarly spread across the seropositivity rate range observed for this sector, not evidencing any testing site-induced bias (see figure below, each dot represents a facility with dot size indicating number of employees participating).

Additionally, we did not see any indications of bias when looking at the overall seropositivity rate across testing sites (see table below).

Testing site	N	Seropositive n (%)
Testing site 1	3065	280 (9.1)
Testing site 2	2186	235 (10.8)
Testing site 3	3004	276 (9.2)
On-site testing	2258	235 (10.4)

We have added the following sentences to **Supplementary Table 1** (last section):

“Study Participation

Three testing centers were selected for the SEROCov-WORK+ study, distantly located across the main urban area of the Geneva canton to allow most employees to find one study testing center at a reasonable distance from their home or work address. Participants came to one of these testing centers for an interview and blood collection appointment that lasted 20 minutes. [...] In the case of some facilities, after identification of critical constraints associated to their participation at the testing centers, the study testing team (constituted of research assistants and nurses) went itself to the premises of the facility to allow the employees to participate in the study directly at their working place. The procedures followed in these cases were the same as those at the testing center. We found no indication of testing site-induced bias after data collection (see Supplementary Table 7).”

The supplemental material includes a description of the survey completed by all participated at time of blood draw. This survey included “risk behaviors”. Please describe these risk behaviors. Were they included in the analysis?

Our response: We thank the Reviewer for this comment. The risk behaviors measured in our survey included only cigarette and e-cigarette smoking behavior. To clarify this, we have rephrased the relevant sentence as follows (Supplementary Table 1):

*“Participants signed a free and informed consent form, and completed a questionnaire with questions related to their employment, work-related activities and preventive measures during the spring 2020 lockdown, **smoking behavior**, and history of comorbidities and COVID-19-related symptoms.”*

As shown in Table 1, we included smoking in the relative risk analyses. We found that smoking was associated with a lower risk of infection, reflecting findings reported elsewhere, including in our population-based survey.⁵

Was exposure to a person with known COVID-19, particularly outside of the work place, asked in the questionnaire? The findings that those “confined” participants had a higher observed risk of being seropositive is evidence that this household exposure is important.

Our response: We thank the Reviewer for this comment. Indeed, the questionnaire collected information about exposure to confirmed COVID-19 cases outside the workplace. As mentioned above, we have now included this information in the main table (Table 1) and in supplementary material (Supplementary tables 5-6). The seropositivity rate among participants who reported being exposed to at least one confirmed COVID-19 case outside of the work place tended to be around twice as high as among participants in the same sector or occupation category who reported no confirmed out-of-work exposure. At the same time, when looking at seropositivity by sectors only among participants with no reported out-of-work exposure to confirmed cases, the pattern of variation in seropositivity rates across sectors and occupations remained similar to that observed in the main results.

We have now rephrased the relevant section in the results to reflect this (page 5, paragraph 1 and 3, respectively):

*“Participants who were fully confined during the lockdown were more likely to be seropositive (1.27; 95% CrI: 1.03-1.56), compared with mobilized participants. **In line with this finding, participants with at least one out-of-work exposure to a confirmed COVID-19 case were more likely to be seropositive (2.29; 95% CrI: 1.93-2.74) compared with their counterparts without any exposure.**”*

“Disaggregating seropositivity rates by out-of-work exposure showed that participants with at least one out-of-work exposure to a confirmed COVID-19 case generally had twice the seropositivity rate compared with their same-sector or same-occupation counterparts with no out-of-work exposure (Supplementary tables 5-6).”

We have also added the following to the discussion (page 7, paragraph 1):

“Participants with out-of-work exposure to confirmed cases of COVID-19 had twice the risk of infection compared with their non-exposed counterparts in the same sector and occupation. This out-of-work exposure is likely determined by several socioeconomic and demographic factors, which in turn are also associated with occupation. While our results showed that out-of-work exposure was associated with higher seropositivity rates, it only partly explained the wide heterogeneity observed across sectors and occupations.”

Discussion

Limitation section should be expanded to include discussion of the possibility of exposure outside of the work place and that this exposure may vary by job type and demographic factors.

Our response: We thank the Reviewer for this suggestion. We agree and, as mentioned above, we have now added the following sentence to the discussion (page 7, paragraph 1):

“Participants with out-of-work exposure to confirmed cases of COVID-19 had twice the risk of infection compared with their non-exposed counterparts in the same sector and occupation. This out-of-work exposure is likely determined by several socioeconomic and demographic factors, which in turn are also associated with occupation. While our results showed that out-of-work exposure was associated with higher seropositivity rates, it only partly explained the wide heterogeneity observed across sectors and occupations.”

Limitations section should also mention that the period of exposure (early in the pandemic) included periods with and without mitigation strategies such as social distancing and masking making it difficult to understand what occurred before or after these interventions.

Our response: We agree, and have now expanded the limitations section as follows (page 7, paragraph 2):

“Importantly, the time frame captured in our data included periods before and after population-wide government-mandated preventive measures were implemented—including heterogeneous sector-specific measures and a lockdown, followed by progressive lifting of lockdown measures. It is likely that the inter- and intra-sectoral variability in seropositivity rate could be partly explained by preventive measures. However, our data do not allow to account for the potential impact of preventive measures.”

References

1. République et Canton de Genève. COVID19 à Genève. Données cantonales. Available at: <https://infocovid.smc.unige.ch/>. (Accessed: 9th March 2021)
2. Clouston, S. A. P., Natale, G. & Link, B. G. Socioeconomic inequalities in the spread of coronavirus-19 in the United States: A examination of the emergence of social inequalities. *Soc. Sci. Med.* 1982 **268**, 113554–113554 (2021).
3. Marí-Dell’Olmo, M. *et al.* Socioeconomic Inequalities in COVID-19 in a European Urban Area: Two Waves, Two Patterns. *Int. J. Environ. Res. Public Health* **18**, 1256 (2021).
4. Wachtler, B. *et al.* Socioeconomic inequalities and COVID-19 – A review of the current international literature. *J. Health Monit.* **5**, (2020).
5. Richard, A. *et al.* Seroprevalence of anti-SARS-CoV-2 IgG antibodies, risk factors for infection and associated symptoms in Geneva, Switzerland: a population-based study. *medRxiv* 2020.12.16.20248180 (2020). doi:10.1101/2020.12.16.20248180

REVIEWERS' COMMENTS

Reviewer #1 (Remarks to the Author):

The revision has improved the clarity of the manuscript. The findings are of importance and merit publication. I have nothing further to add.

Reviewer #2 (Remarks to the Author):

Thank you for successfully addressing the comments and questions from the initial review. The addition of data of the RR for SARS-CoV-2 seropositivity of roughly 2 for those exposed to confirmed COVID-19 cases out of work is important. Given the magnitude of this risk, out-of-work exposure should be incorporated into the risk adjustment model as it could be a major confounder.

The inability to account for all out of work exposures should also be included in the limitations paragraph

Reviewer #3 (Remarks to the Author):

Note that I am providing a second-round review for this manuscript, having not been involved in the first round. I was specifically asked to comment on the statistical model.

This is a useful, concisely reported dataset comparing anti-spike IgG seropositivity rates in a single Swiss canton (Geneva) across occupations. The authors found substantial heterogeneity in seropositivity rates within sectors and participating facilities, but no evidence of systematically higher risk in so-called "high-risk" occupations such as nursing home workers, despite high seropositivity rates in some facilities. Other notable increased or decreased risk associations were described, for example, associations between education level or out-of-work exposure and seropositivity.

The statistical model is well described and seems appropriately defined. Sensitivity and specificity are taken into account and I like the way the lab validation data are included directly in the model. The choice of priors for the regression coefficients seems sensible (not overly flat, which can lead to problematic prior predictive distributions in logistic regression). The hierarchical model for company-specific random effects grouped by sector seems sensible. Convergence assessment was acknowledged which is reassuring.

*** Comments ***

- The authors present two logistic regression models in the supplement: one including intercept terms for facility (called company in the supplement) and one without. It is not immediately clear which model was used for which results. Were the relative risks in the main text for eg. smoking status, education, based on the first model or after accounting for facility and sector? Consider making it clearer which model was used for which results.
- It wouldn't be of interest to most readers, but some more posterior prediction plots or posterior densities for the relative risks and regression coefficients might be a useful addition to the supplement. For example, there are a few pointrange plots for the raw data, but nothing for the posterior estimates. I leave that to the authors discretion though.
- The main text method states "random effects for sectors and/or facilities". When were the intercept terms defined by sector rather than facility? Consider rewording to be more precise.
- Given how easy it is to post Stan/R code on GitHub, I would suggest that the authors make the analytical code publicly available rather than "available on request". My review of method implementation is limited to what is written down in the supplement, so I cannot assess if the code is implemented correctly or reproducibly.

*** Minor comments ***

- Is there any particular reason why the priors were not included in the model equations in the supplement, but only stated in the text? Might be easier to just include them.

- Very minor question, but are the reported risk ratio point estimates posterior medians, medians or other? It looks like this is posterior mean according to the supplement. Consider stating in main text.
- Participants who were fully confined during the lockdown were more likely to be seropositive (1.27; 95% CrI: 1.03-1.56). Missing "RR: 1.27" as in the other brackets. Same with final sentence of this paragraph.
- Reference 32 is missing.
- It's kind of odd that the supplementary methods are embedded as an image file – can this be made part of the word document?
- Table 1 shows that being a smoker or ex-smoker is associated with a significantly lower risk of seropositivity relative to being a non-smoker. Can the authors speculate on why this is? Is this linked to the confounding of smoking status with occupation (similar to education and occupation)? I see that this is mentioned by the authors in response to Reviewer 2. Perhaps make this link and relation to previous results clear in the discussion.
- Did the authors consider including date of survey or phase of canton interventions in the model? Perhaps they were underpowered to do so.
- Reviewer 1 highlighted that the heterogeneity within sectors could be interesting (to which the authors responded well). Are there any interesting findings in the posterior estimates for the sector-specific variances? Presumably not due to the confounders described by the authors in their response.

James Hay

Response to reviewers' comments

Reviewer #1

The revision has improved the clarity of the manuscript. The findings are of importance and merit publication. I have nothing further to add.

Our response: We thank the reviewer for this encouraging comment and for their contribution to improving our manuscript.

Reviewer #2

Thank you for successfully addressing the comments and questions from the initial review. The addition of data of the RR for SARS-CoV-2 seropositivity of roughly 2 for those exposed to confirmed COVID-19 cases out of work is important. Given the magnitude of this risk, out-of-work exposure should be incorporated into the risk adjustment model as it could be a major confounder.

Our response: We are grateful for the reviewer's comments. We have now further adjusted the relative risk of being seropositive for out-of-work exposure and found that the results remained virtually the same. We have added these results as supplementary table 7. We have added the following sentence in the discussion (page 5, paragraph 3):

“Further adjusting relative risk estimates for known out-of-work exposures, however, showed no appreciable change from the main results.”

The inability to account for all out of work exposures should also be included in the limitations paragraph.

Our response: We have now added the following sentence to the limitations (page 7, paragraph 2):

“Finally, although we accounted for self-reported out-of-work exposure to confirmed cases of COVID-19, unknown exposure to SARS-CoV-2 outside the work setting, such as in transportation to and from work, may have also played a role.”

Reviewer #3

Note that I am providing a second-round review for this manuscript, having not been involved in the first round. I was specifically asked to comment on the statistical model.

This is a useful, concisely reported dataset comparing anti-spike IgG seropositivity rates in a single Swiss canton (Geneva) across occupations. The authors found substantial heterogeneity in seropositivity rates within sectors and participating facilities, but no evidence of systematically higher risk in so-called “high-risk” occupations such as nursing home workers, despite high seropositivity rates in some facilities. Other notable increased or decreased risk associations were described, for example, associations between education level or out-of-work exposure and seropositivity.

The statistical model is well described and seems appropriately defined. Sensitivity and specificity are taken into account and I like the way the lab validation data are included directly in the model. The choice of priors for the regression coefficients seems sensible (not overly flat, which can lead to problematic prior predictive distributions in logistic regression). The hierarchical model for company-specific random effects grouped by sector seems sensible. Convergence assessment was acknowledged which is reassuring.

Our response: We thank the reviewer for these encouraging comments.

*** Comments ***

- The authors present two logistic regression models in the supplement: one including intercept terms for facility (called company in the supplement) and one without. It is not immediately clear which model was used for which results. Were the relative risks in the main text for eg. smoking status, education, based on the first model or after accounting for facility and sector? Consider making it clearer which model was used for which results.

Our response: We thank the reviewer for these comments. We have now replaced company for facility in the supplement, and have specified which model was used for which results (page 5 of the supplement):

“The above model was used to estimate all the relative risks presented except for those pertaining to the activity sectors.”

- It wouldn't be of interest to most readers, but some more posterior prediction plots or posterior densities for the relative risks and regression coefficients might be a useful addition to the supplement. For example, there are a few pointrange plots for the raw data, but nothing for the posterior estimates. I leave that to the authors discretion though.

Our response: We thank the reviewer for this suggestion. We agree, and have now added the following figures to the supplement:

Supplementary figure 5. Posterior densities for sector-specific relative risk estimates

Magenta line is the reference posterior median from healthcare sector.

Supplementary figure 6. Posterior distributions of sector-specific mean estimates

Black vertical lines in each sector indicate the quartiles of the estimated densities.

Supplementary figure 7. Posterior distributions of sector-specific variance estimates

Black vertical lines in each sector indicate the quartiles of the estimated densities.

- The main text method states “random effects for sectors and/or facilities”. When were the intercept terms defined by sector rather than facility? Consider rewording to be more precise.

Our response: We have now rephrased the corresponding sentence to clarify this:

“... we fit Bayesian regression models that accounted for age, sex, and test performance including, when appropriate, random effects for facilities.”

- Given how easy it is to post Stan/R code on GitHub, I would suggest that the authors make the analytical code publicly available rather than “available on request”. My review of method implementation is limited to what is written down in the supplement, so I cannot assess if the code is implemented correctly or reproducibly.

Our response: We thank the reviewer for this suggestion. We agree and have now made our analytical code available. The relevant code availability sentence now reads (page 9):

“Analytical code is available from <https://github.com/UEP-HUG/SEROCO-V-WORK-Public>.”

*** Minor comments ***

- Is there any particular reason why the priors were not included in the model equations in the supplement, but only stated in the text? Might be easier to just include them.

Our response: We thank the reviewer for pointing this out. We have now also included the priors in the model equations in the supplement.

- Very minor question, but are the reported risk ratio point estimates posterior medians, means or other? It looks like this is posterior mean according to the supplement. Consider stating in main text.

Our response: We thank the reviewer for this question. The reported estimates are posterior means. We have now clarified the corresponding sentence in the methods (page 9, paragraph 1):

“To estimate the relative risk (posterior mean and 95% central credible interval)...”

- Participants who were fully confined during the lockdown were more likely to be seropositive (1.27; 95% CrI: 1.03-1.56). Missing “RR: 1.27” as in the other brackets. Same with final sentence of this paragraph.

Our response: We thank the reviewer for pointing this out. We have corrected these sentences:

“Participants who were fully confined during the lockdown were more likely to be seropositive (RR: 1.27; 95% CrI: 1.03-1.56), compared with mobilized participants. In line with this finding, participants with at least one out-of-work exposure to a confirmed COVID-19 case were more likely to be seropositive (RR: 2.29; 95% CrI: 1.93-2.74) compared with their counterparts without any exposure.”

- Reference 32 is missing.

Our response: We thank the reviewer for catching this. We have now fixed this error.

- It’s kind of odd that the supplementary methods are embedded as an image file – can this be made part of the word document?

Our response: We have now added the supplementary methods to the word document.

- Table 1 shows that being a smoker or ex-smoker is associated with a significantly lower risk of seropositivity relative to being a non-smoker. Can the authors speculate on why this is? Is this linked to the confounding of smoking status with occupation (similar to education and occupation)? I see that this is mentioned by the authors in response to Reviewer 2. Perhaps make this link and relation to previous results clear in the discussion.

Our response: We thank the reviewer for this comment. We have now added the following sentence in the discussion (page 7, paragraph 1):

“The observed reduced risk of being seropositive in ex-smokers and current smokers reflected our previous population-based findings;²¹ this may be due to potential interaction between the SARS-CoV-2 spike protein and nicotinic acetylcholine receptors³⁴, heightened preventive practices and risk aversion among smokers given their increased risk of viral and bacterial respiratory diseases,^{32,33} ³⁴ or residual confounding. Importantly, while contrasting findings have been reported about the association between smoking and SARS-CoV-2 infection risk, extensive evidence indicates that smoking is associated with increased risk of severe COVID-19.^{35,36}”

- Did the authors consider including date of survey or phase of canton interventions in the model? Perhaps they were underpowered to do so.

Our response: We thank the reviewer for this question. Indeed, we decided against including these data into the model due to insufficient power, in addition to the complexity of conducting such analysis. As we explained in the previous response letter, the type, duration, and strictness of preventive measures varied widely across sectors and facilities before the government-mandated lockdown measures were implemented. Additionally, the level of adherence and implementation to government-mandated measures across facilities remains uncertain.

- Reviewer 1 highlighted that the heterogeneity within sectors could be interesting (to which the authors responded well). Are there any interesting findings in the posterior estimates for the sector-specific variances? Presumably not due to the confounders described by the authors in their response.

Our response: We thank the reviewer for this question. We now show the posterior distributions of sector-specific variance estimates in supplementary figure 7—we do not observe any noteworthy or unexpected estimates.